# Characterization of Human Papillomavirus 16 from Kinshasa (Democratic Republic of the Congo)—Implications for Pathogenicity and Vaccine Effectiveness

**DOI:** 10.3390/microorganisms10122492

**Published:** 2022-12-16

**Authors:** Paula Iglesias, Celine Tendobi, Silvia Carlos, Maria D. Lozano, David Barquín, Luis Chiva, Gabriel Reina

**Affiliations:** 1Microbiology Department, Clínica Universidad de Navarra, 31008 Pamplona, Spain; 2Department of Obstetrics and Gynecology, Centre Hospitalier Mère-Enfant (CHME), Ngafani, Kinshasa 4484, Democratic Republic of the Congo; 3Department of Preventive Medicine and Public Health, Universidad de Navarra, 31008 Pamplona, Spain; 4IdiSNA, Navarra Institute for Health Research, 31008 Pamplona, Spain; 5ISTUN, Institute of Tropical Health, Universidad de Navarra, 31008 Pamplona, Spain; 6Department of Pathology, Clínica Universidad de Navarra, 31008 Pamplona, Spain; 7Department of Obstetrics and Gynecology, Clínica Universidad de Navarra, 28027 Madrid, Spain

**Keywords:** Human Papillomavirus 16, Kinshasa, L1, LCR, variants, cervical cancer

## Abstract

Human Papillomavirus (HPV) type 16 is the main etiological agent of cervical cancer worldwide. Mutations within the virus genome may lead to an increased risk of cancer development and decreased vaccine response, but there is a lack of information about strains circulating in Sub-Saharan Africa. Endocervical cytology samples were collected from 480 women attending a voluntary cervical cancer screening program at Monkole Hospital and four outpatient centers in Kinshasa, Democratic Republic of the Congo (DRC). The prevalence of HPV infection was 18.8% and the most prevalent high-risk types were HPV16 (12.2%) followed by HPV52 (8.8%) and HPV33/HPV35 (7.8% each). HPV16 strains were characterized: 57.1% were classified as C lineage; two samples (28.6%) as A1 and one sample belonged to B1 lineage. HPV33, HPV35, HPV16, and HPV58 were the most frequent types associated with low-grade intraepithelial lesion while high-grade squamous intraepithelial lesions were predominantly associated with HPV16. Several L1 mutations (T266A, S282P, T353P, and N181T) were common in Kinshasa, and their potential effect on vaccine-induced neutralization, especially the presence of S282P, should be further investigated. Long control region (LCR) variability was high with frequent mutations like G7193T, G7521A, and G145T that could promote malignancy of these HPV16 strains. This study provides a helpful basis for understanding HPV16 variants circulating in Kinshasa and the potential association between mutations of LCR region and malignancy and of L1 and vaccine activity.

## 1. Introduction

Human Papillomavirus (HPV) infects mucosal and cutaneous epithelia, and can be sexually transmitted [1,2,3,4]. It is described as one of the most common sexually transmitted infections (either skin-to-skin or mucosa-to-mucosa contact). Although mostly asymptomatic and cleared from the body by the immune system, HPV infection can cause disease, the usual manifestations of which are anogenital warts, cutaneous warts, recurrent respiratory papillomatosis, cervical intraepithelial neoplasia (CIN), cervical, vulvar, penile, and anal cancers, and carcinomas of the upper respiratory tract [5,6]. Besides oncogenic HPV activity, cervical cancer onset and development encompass a large variety of genetic and epigenetic changes [7,8]. According to the risk of malignancy, HPV is classified as low-risk (LR-HPV) or high-risk HPV (HR-HPV). An HR-HPV infection is considered a necessary cause for the transformation of the infected epithelia [9,10,11]. HPV16, the main HR-HPV, is categorized as “Group 1 Agent: Carcinogenic to Humans” by the International Agency for Research on Cancer (IARC) because of the strong association between infection and cervical cancer [12], which is a serious public health problem, especially in low- and middle-income countries [13].

The burden of cervical cancer is higher in regions with a low and medium human development index [14]. The estimated global HPV prevalence is 11.7%, with Sub-Saharan Africa (24.0%), Eastern Europe (21.4%), and Latin America (16.1%) showing the highest prevalences [15,16]. Recent data from the Democratic Republic of the Congo (DRC) reported a prevalence of 28.8% in this country [16,17], and cervical cancer ranks as the first cause of cancer in incidence and mortality in DRC among women (second in the general population) [18].

Considering the specific types, the five most prevalent agents worldwide in women with normal cytology and cervical neoplastic diseases are HPV16, 18, 31, 52, and 58 [15,19]. In particular, HPV16 and HPV18 are responsible for approximately 70% of cervical cancers. The prevalence of HPV16 worldwide was estimated to be 4.4% among women with normal cytology, 14.5% with low-grade intraepithelial lesion (LSIL), 31.2% with high-grade squamous intraepithelial lesion (HSIL), and 49.7% with invasive cervical cancer (ICC), but scarce data on women from Central Africa are available [16]. In DRC, a prevalence of 38.6% and 67.2% of HPV16/HPV18 infection among women with HSIL and ICC, respectively, has been reported [16].

HPV structure is described as a double-stranded circular DNA virus with eight protein-coding genes [20] and around eight thousand base pairs that can be divided into three different functional parts: the early region, which encodes proteins for viral replication (E1–E7); the late region, which encodes structural proteins for virion assembly (L1–L2); and a non-coding part, the long control region (LCR), with necessary elements for replication and transcription, which is under epigenetic regulation [12,21]. L1 provides an important role in the infectious interaction between the virus and the host cells. It encodes recombinant L1 proteins created to assemble Virus-Like Particles (VLPs), which are the base of the available vaccines [22]. The LCR is the most variable region of HPV and contains transcriptional regulatory sites for viral and cellular proteins, which may have an important role in the development of cervical cancer [23]. In particular, LCR contains the early promoter p97 (in the E6 proximal part of the region) that controls the regulation of the early E6 and E7 oncogene expression [24]. It also contains the keratinocyte enhancer region, the origin of replication, and the binding sites for various transcription factors and the viral helicase E1 as well as the viral protein E2, which contributes to viral replication and regulation of viral gene expression [21].

The variations in L1 and LCR fragments allow the identification of 6 major lineages of the different HPV types [25]: European (EUR), Asian (AS), Asian-American (AA), North American (NA), African-1 (AFR1), and African-2 (AFR2), based on the origin of the most prevalent variant [26,27]. Alternatively, they can be also classified as the lineages and sublineages A1-4 (EUR-AS), B (AFR1), C (AFR2), and D (NA/AA) [28].

Changes in the genomic sequence of HPV are rare, as the maximal genomic divergence of HPV16 is about 5% [29]. The frequency of mutations differs within the different regions; approximately 35.5% of the L1 region sequences are reported to show mutations, while this number goes up to 75% for the LCR region [30]. However, the information published in databases mainly includes sequences from Europe, America, and Asia, with a lack of information from Africa.

The objective of this study was to analyze the epidemiology of HPV types in Kinshasa and their role in LSIL and HSIL. We also wanted to investigate HPV16 variants circulating in Kinshasa through sequencing of the LCR and L1 regions and to estimate the persistence capacity, risk of cancer development, and response capacity to vaccine-induced neutralization according to genome analysis.

## 2. Materials and Methods

### 2.1. Study Design and Participants

A cross-sectional study was designed to evaluate a screening program of cervical pre-cancerous lesions at Monkole Hospital and four outpatient centers (Kimbondo, Eliba, Moluka, and Gombe) in Kinshasa (DRC). During July 2017, women aged 25–70 years attending HIV Voluntary Counseling and Testing (VCT) or seeking medical care were invited to participate. Pregnant women, those not-sexually initiated, or those with a total hysterectomy were excluded. No patient had previously been diagnosed or treated for HPV disease.

### 2.2. Sample Collection

Four hundred and eighty women were screened for LSIL, including cervical intraepithelial neoplasia grade 1 (CIN1); HSIL, including CIN2/3; and ICC, through visual inspection with acetic acid (VIA) and Lugol according to WHO guidelines [31,32]. In addition, endocervical samples were collected for HPV testing and characterization. Specimens were collected using BD SurePath™ liquid-based Pap test and shipped to Clinica Universidad de Navarra (Pamplona, Spain), where they were stored at 4 °C until further use.

### 2.3. HPV Molecular Detection

HPV DNA detection and genotyping of the different LR-HPV and HR-HPV types was performed using the clinical array technology CLART HPV4 assay (Genomica, Madrid, Spain). This method is able to detect and identify 35 clinically relevant HPV types: HR-HPV (16, 18, 31, 33, 35, 39, 45, 51, 52, 56, 58, 59, 66, 68A-68B); probable HR-HPV (26, 53, 73, 82); and LR-HPV (6, 11, 40, 42, 43, 44, 54, 61, 62, 71, 70, 72, 81, 83, 84, 85, 89) [33]. Subsequently, L1 and LCR regions from HPV16-infected patients were sequenced for lineage determination and mutations analysis through an in-house amplification protocol followed by Sanger sequencing.

### 2.4. DNA Extraction for HPV16 Characterizatin

DNA was manually extracted using 200 µL of endocervical sample medium with High-Pure Viral Nucleic Acid (Roche) kit following the manufacturer instructions. Briefly, 253 μL of Lysis buffer was prepared (250 μL Binding buffer + 3 μL Poly A carrier DNA) and together with 50 μL of Proteinase K was added to the endocervical sample and incubated 10 min at 72 °C. After a short centrifugation (9300 rpm for 5 s), 100 μL of Binding buffer were added to the former mix. Then, DNA purification was performed using the manufacturer filter columns and centrifugation (9300 rpm for 1 min) after each step. First, 500 μL of Inhibitor Removal Buffer was used for purification and two additional steps with 450 μL of Wash Buffer were carried out. Lastly, for DNA elution 50 μL of Elution Buffer was used and after centrifugation at 9300 rpm/1 min, the purified DNA was stored at −70 °C until use.

### 2.5. Primers Design and HPV16 Amplification

The L1 gene was sequenced following the amplification of two fragments of 911 bp (L1.1) and 879 bp (L1.2), to finally obtain the L1 complete sequence of 1498 bp (amino acids 1–499; reference sequence NC_001526.4: 4775-6292). Similarly, LCR was divided in two fragments of 460 bp (LCR.1) and 621 bp (LCR.2) for a final complete LCR sequence of 985 bp (reference sequence NC_001526.3).

Primers previously described for L1 [34] and LCR [23] regions and new designed primers after studying all HPV16 lineages (including all known African variants) were used for amplification (Table 1).

PCR reactions were performed using 5 μL of DNA for each target gene in a final volume of 25 μL reaction mix containing 1.5 pmol of each forward and reverse primers (for L1.1. and L1.2. reactions) or 2 pmol of each forward and reverse primers (for LCR.1. and LCR.2. reactions), 12.5 μL KAPA2G Fast HotStart ReadyMix (2X) (Sigma-Aldrich, Wilmington, MA, USA) and water. All amplifications were carried out a t 95 °C for 5 min and 40 cycles of denaturation at 95 °C for 15 s, annealing at 55 °C for 15 s, extension at 72 °C for 30 s, and a final elongation step at 72 °C for 1 min.

### 2.6. Sequencing and Interpretation

PCR amplicons were purified using the GFX™ PCR DNA and Gel Band Purification Kit (GE Healthcare) and sequenced by CIMA Lab Diagnostics (Navarra, Spain). Sequences were edited with FinchTV and Clustal Omega Multiple Sequence Alignment Website to obtain FASTA files. Then, lineages and mutational sites were identified. All sequences were aligned with HPV16 reference sequences belonging to different lineages (A: NC_001526, AF536179, AY686584, HQ644234, HQ644236, HQ644245, KC150026, KP161041, KU053900, LC456192, LC456606, LC456622, LC456201, MH921945; B: AF472508, AF536180, HQ644298, KF466573, KU053915; C: AF472509, HQ644249, KF466622; D: AF402678, AY686579, HQ644257, HQ644270, KF447501, KF447525, KU053929, KU053943, LC456618) using Clustal Omega Multiple Sequence Alignment Website and MEGAX software to construct phylogenetic trees of HPV 16 L1 and LCR by the Neighbor-Joining method. Finally, nucleotide and amino acid mutations were located using MEGAX software [35,36].

Accession Numbers: L1 and LCR HPV16 sequences were submitted to GenBank (www.ncbi.nlm.nih.gov/genbank, accessed on 5 December 2022) with the following accession numbers: OQ029514-OQ029524.

### 2.7. Ethics

The project was approved by the Human Subjects Review Committees at Monkole Hospital/CEFA (Kinshasa, DRC) and University of Navarra (Pamplona, Spain). Informed consent of enrolled participants was obtained. All methods were carried out in accordance with relevant guidelines and regulations.

## 3. Results

The mean (SD) age of our population was 44.6 (10.9) years old. Ninety participants within four hundred and eighty screened women were infected by HPV (18.8%), 75% of them HR-HPV. There were no differences in age between patients infected and not infected with HPV. Among those infected, 11 out of 90 women were infected by more than one strain (up to seven different types). The most prevalent HR-HPV type was HPV16 found in 11 women (12.2%) followed by HPV52 (8.9%) and HPV33/HPV35 (7.8% each) (Figure 1). 

The distribution of HR-HPV among participants showing LSIL (CIN-1) is shown in Figure 2. HPV33 (17.1%) and HPV35 (14.6%) were the most frequent types associated with LSIL, followed by HPV16 and HPV58 (9.8%). Among participants with HSIL (CIN-2/3), HPV16 was the predominant agent associated with this phase (55.6%) (Figure 3).

All HPV16 positive samples (eleven) were subjected to L1 and LCR sequencing. Six out of eleven (54%) and eight out of eleven (72%) samples were positive after L1 and LCR amplification, respectively. However, after the sequencing reaction, three sequences had to be discarded due to low quality, so the final number of available sequences was five L1 sequences and six LCR sequences, belonging to seven different patients.

### 3.1. HPV16 Lineage Distribution

Phylogenetic analysis of both regions (L1 and LCR) allowed us to obtain the lineage of seven out of eleven HPV16 infections detected among the screened women. The African lineage AFR2 was the most prevalent within this cohort, followed by the European ones. Four samples (57.1%) were classified as C lineage; two samples (28.6%) as A1; and one sample belonged to B1 lineage (Figure 4 and Figure 5).

### 3.2. L1 Genomic Polymorphisms

Seventeen single nucleotide changes were identified among the five L1 sequences obtained. These mutations included 10/17 (58.8%) silent mutations, and 7/17 (41.2%) which led to amino acids substitutions (Table 2). T266A substitution was present in 80% of all the samples studied for this fragment; then S282P and T353P were present in 3/5 sequences; and L474F in two out of five samples. Finally, H76Y, T176N, and N181T were identified in one patient; these changes are located at the first segment of L1 region, only obtained for two samples (P44 and P114). 

### 3.3. LCR Genomic Polymorphisms

Thirty-five single nucleotide changes were identified among six LCR sequences (Table 3). The most prevalent nucleotide mutations were G7193T, G7489A, G7521A, T7834G, and T31C. Additionally, C7485A, C7669T, G7826A, A7837C, and T109C were present in three out of six samples, all belonging to lineage C (AFR2). 

In addition, two consecutive positions showed nucleotide changes in lineage C: G7435A alone (highly frequent) or combined with G7436C.

Overall, 52 single nucleotide changes (17 in L1 and 35 in LCR) were identified, which led to 10 amino acid changes (10 in L1 and none in LCR as it is a non-coding region) among HPV16 strains circulating in Kinshasa (DRC).

## 4. Discussion

This study provides the first data on the lineages of HPV16 from Kinshasa (DRC) and constitutes a helpful experimental basis for understanding HPV16 variants in this area, the most prevalent L1 and LCR mutations, and the correlation between mutations of the LCR region and malignancy and L1 region and immunogenicity of the virus. To our knowledge, there are no previous studies that have included a full description of L1 and LCR from HPV16 mutations and variants collected in DRC.

The prevalence of HPV infection within our cohort was 18.8%, similar to that reported before in Kinshasa between 11–28% [17,39]. However, the main HR type found among our participants was HPV16 (12.2%), different from the distribution reported before where HPV68 and HPV35 were the most prevalent HR types detected [17,39].

Considering the presence of HPV types among our participants showing intraepithelial lesions, HPV33 and HPV35 were the most frequent types associated with LSIL, followed by HPV16 and HPV58, while HPV16 was the main virus associated with HSIL. Like our study, previous data from Kinshasa have highlighted an important incidence of HPV35, HPV52, HPV16, and HPV18 among women with LSIL and HSIL; however, HPV68, previously reported, was not predominant in our series [40]. Collected data from Eastern, Southern, and Western Africa stated, like our study, that HPV16, HPV18, HPV35. and HPV52 were the leading types associated with LSIL [19]. The information from Central Africa is limited; we found also a high presence of HPV33 and HPV58 among women with LSIL, uncommon strains in other areas of the continent among LSIL. HSIL were mainly associated in our series with HPV16, as described in other African countries. Other HPV types were also found in HSIL participants where HPV39 has not been found associated with this phase in other countries where data is available [19].

Overall, L1 looks to be a more conserved region, as shown in previous studies, with 17 single nucleotide changes found, versus 35 mutations detected in LCR. Furthermore, mutations in L1 (when they occurred) were present in almost all samples, unlike LCR, where different patterns were found with changes present in one single sample. This may be due to the different functions of each gene, with L1 being a part of the virus assembly, while LCR is involved in regulating the expression of different proteins.

L1 plays a vital role in the efficacy of vaccines against HPV16 that are currently on the market. This major capsid protein sequence is usually well conserved, but mutations can have an impact in the virus sensitivity to antibodies induced by vaccines, especially when they are present on the immune dominant loops [41]. Our samples from Kinshasa showed some mutations related to this interaction, like A797G, T844C, A1057C, and A542C mutations (amino acid substitutions T266A, S282P, T353P, and N181T, respectively).

Ning et al. studied how the changes in the amino acid sequence of the L1 region of HPV16 influenced the susceptibility to antibody-mediated neutralization by creating pseudovirions. Their study showed that amino acid positions 181 (EF loop) or 282 (FG loop) were critical residues involved in the conformational epitopes recognized by monoclonal antibodies (MAbs). Especially the S282P mutation induced decreased reactivity to the virus for 7/13 MAbs studied and could lead to a complete loss of susceptibility to neutralization [42]. In other studies, T266A (FG loop) and T353P (HI loop) were reported to destabilize the L1 monomer structure due to loss of hydrogen bonds caused by those amino acid substitutions [43]. They have also been reported to have a potential impact on antigenicity and recognition by antibodies and T cells [42,44].

Overall, H76Y, T176N, N181T, A266T, T353P, and L474F, all mutations present in our samples from Kinshasa, have been described as variations in immune dominant loops of HPV16 L1, and might affect the efficacy of the vaccines available [45]. These mutations have been previously reported in different variants, the most common being T353P, in 20.1% of samples, followed by H76Y (15.3%), within collections including little information from Africa [42].

Further research needs to be done, because scarce data on L1 mutation prevalence when talking about African lineages are available [28,45,46]. For the rest of the lineages, some of these mutations have shown ethnicity differences; T176N mutation is more prevalent between Asian and American populations (23.4–25.3%), while A266T is more prevalent in the European population (16.4%) [45].

LCR has been described as the most variable region of the HPV16 genome [47,48,49], and it is well known for its role in the development of cervical cancer. Changes in this region can influence not only the transcription of viral oncogenes but the addition/loss of binding sites for transcription factors [23,50]. Nowadays, there are plenty of studies that link the presence of certain mutations with the malignity of HPV16, but the actual role of the specific SNPs and their link to transcriptional factors, as well as the actual role they play in the likelihood of cancer development, are not well established. 

Our data show mainly the presence of G7193T, G7489A, G7521A, T7834G, and T31C mutations (amino acid substitutions C2398F, V2497I, M2507I, V2612L, and R11C, respectively) among Kinshasa patients. In our study, G7521A was the most frequently observed mutation in LCR (83%) in concordance with other studies [23,26,30,51] and was highly distributed in patients with cervical cancer. 

Some studies locate this mutation at the binding site of the transcriptional repressor sex-determining region Y-box 9 protein (SOX9) [23]. This could down-regulate the expression of this gene, which is thought to act as a tumor suppressor in the development of cervical cancer [52].

On the other hand, other studies describe this mutation as involved in the YY1 binding site, a mutation that is found in the majority of patients with cervical cancer, and that may promote the activity of p97, involved in the transcription of other oncogenes. Both mechanisms are believed to have a role in the development of tumors for different cancers [30,53,54].

Other mutations described in the literature, such as G7193T, have been linked to the transcriptional enhancer factor (TEF-1) [51,54] that influences the regulation of the LCR transcription activity and, like YY1, acts as a critical activator protein for p97 activity [55]. Furthermore, the G145T mutation is described in other studies, suspected to affect p53 binding and degradation as well as acting like a B and T cell epitope [44].

According to phylogenetic analysis of L1 and LCR sequences, HPV16 strains detected in our samples from Kinshasa belonged mostly to lineage C (AFR2) (57%), but strains belonging to lineage A1 and B were also found. Cornet et al. described a number of E2/LCR genetic variations in HPV16 that correlate with phylogenetic classification. In this sense, mutations such as C7485A, C7669T, C143G, and G145T were present in samples classified as African lineage. In addition, strains classified as lineage C (AFR2) contained specific mutations such as G7826A, A7837C, G7387C, and G7435A. One sample classified as lineage B1 could be assigned to sublineage AFR1a because of the presence of T7714A and T7293G [27]. Finally, two samples were classified as A1 (EUR) variants and they did not show any of the classic AFR mutations described, but were more similar to the reference sequence.

Our study is subject to a number of potential limitations. The first is that only HPV16 was studied and the sample size of HPV16 detected and sequenced was low; however, the information obtained from L1 and LCR sequences was homogenous and consistent to be able to analyze the genetic polymorphisms of HPV16 in this area. Second, no information about HPV vaccination was obtained from our patients, but unfortunately, a vaccination campaign against HPV has not been initiated in DRC, so no effect from vaccination could be expected. Introducing inexperienced medical providers into the cervical cancer screening campaign may have had a potential bias. However, this suggests that these campaigns might be feasible in the context of amateur observers after short training. Finally, no next-generation sequencing was performed; however, Sanger sequencing is accepted as a reliable tool for HPV typing and polymorphisms detection. Additional sequencing studies should be performed with other HPV types highly prevalent in the DRC, such as HPV18, 33, 35, 52, 53, 61, and 68, so that information obtained from them could be representative of the HPV impact in the region.

The study has several notable strengths, including the description of HPV epidemiology among women from DRC showing LSIL and HSIL using a reliable commercial test. This study described for the first time the distribution of HPV16 L1 and LCR genetic variants and polymorphisms in infected women from the Democratic Republic of the Congo. The characterization of HPV16, using molecular and phylogenetic techniques, from subjects in Kinshasa belonging mainly to the general population, confirmed lineage C (AFR2) as the predominant strain in our study population. This work can help to understand the natural history and carcinogenicity of HPV16 genetic variants, as well as enabling the study of potential vaccine escape mutants in Kinshasa. 

Finally, further analysis with a larger sample size needs to be performed in order to obtain reliable information about all the HPV16 variants circulating in Kinshasa. As previously reported in other African countries, the genomic diversity of HPV16 variants has been demonstrated and the L1/LCR mutations identified should be further investigated using functional assays [28,46]. This will allow future researchers to assess if the use of prophylactic vaccination will protect this population against the virus, and to find possible mutations in these variants that increase the risk of development of cervical cancer.

## 5. Conclusions

In conclusion, the prevalence of HPV infection within women from Kinshasa was 18.8%, with HPV16 as the most common HR type found (12.2%), mainly belonging to lineage C (AFR2). HPV33, HPV35, HPV16, and HPV58 were the most frequent types associated with LSIL, while HSIL were predominantly associated with HPV16. L1 mutations (T266A, S282P, T353P, and N181T) were common among the HPV16 strains detected in Kinshasa, and their potential effect on vaccine-induced neutralization, especially the presence of S282P, should be further investigated. LCR variability of HPV16 in this area was very high with frequent mutations like G7193T, G7521A, and G145T that could promote malignancy.

## Figures and Tables

**Figure 1 microorganisms-10-02492-f001:**
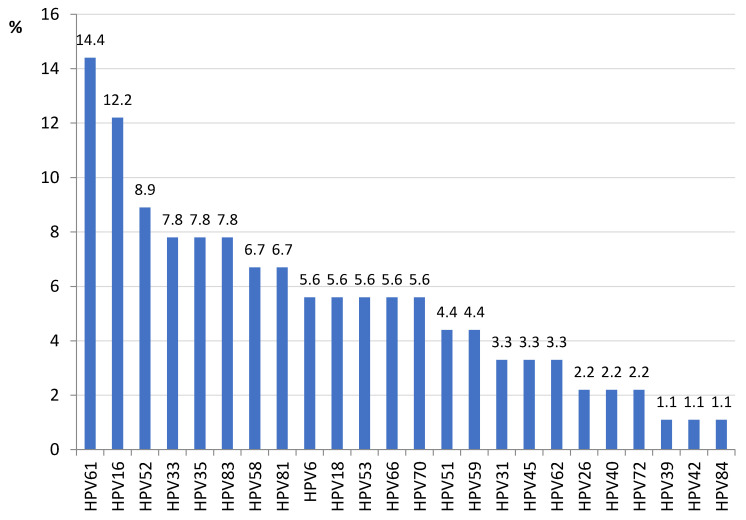
HPV types infecting screened women in Kinshasa (DRC) (n = 90).

**Figure 2 microorganisms-10-02492-f002:**
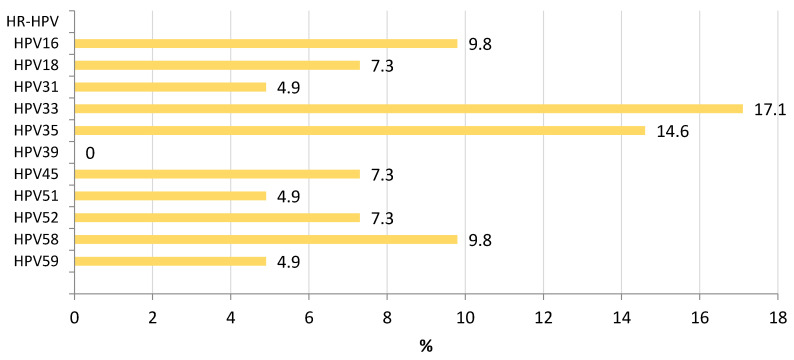
Prevalence of HR-HPV among participants with low-grade intraepithelial lesion (LSIL) in Kinshasa.

**Figure 3 microorganisms-10-02492-f003:**
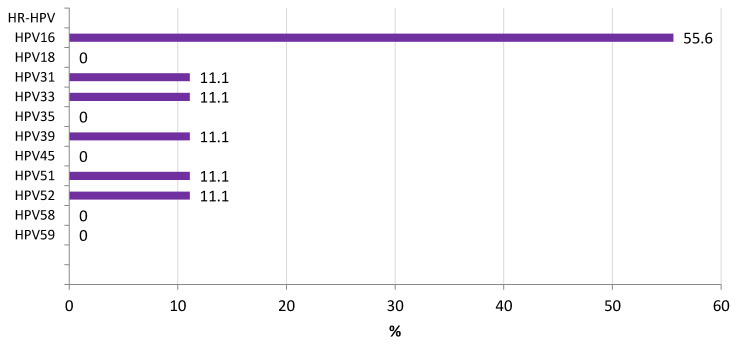
Prevalence of HR-HPV among participants with high-grade squamous intraepithelial lesion (HSIL) in Kinshasa.

**Figure 4 microorganisms-10-02492-f004:**
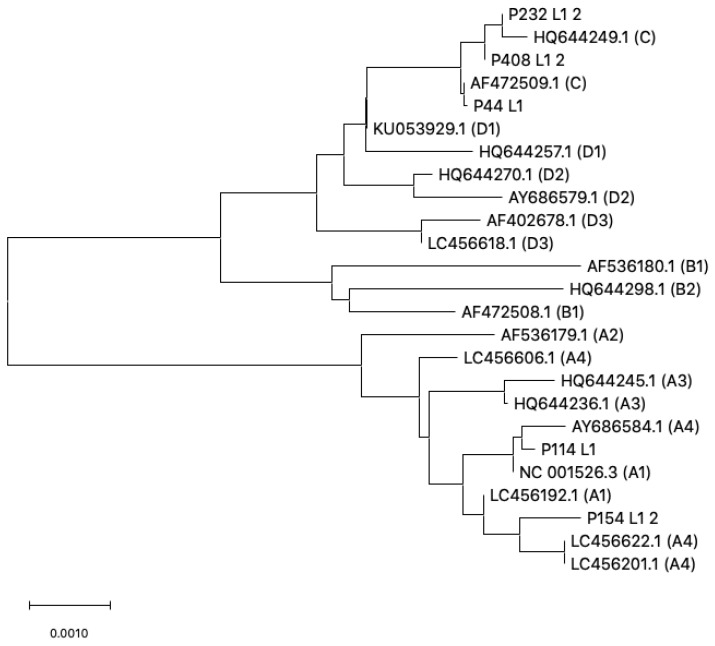
**Evolutionary relationships of HPV16 lineages/sublineages in Kinshasa based on L1 region.** The evolutionary history was inferred using the Neighbor-Joining method [37]. The tree is drawn to scale, with branch lengths in the same units as those of the evolutionary distances used to infer the phylogenetic tree. The evolutionary distances were computed using the Maximum Composite Likelihood method [38] and are in the units of the number of base substitutions per site. The analysis involved 25 nucleotide sequences (20 reference strains from GenBank and 5 new sequences from this study). Codon positions included were 1st + 2nd + 3rd + Noncoding. All positions containing gaps and missing data were eliminated. There were a total of 1518 positions in the final dataset. Evolutionary analyses were conducted in MEGAX [35,36].

**Figure 5 microorganisms-10-02492-f005:**
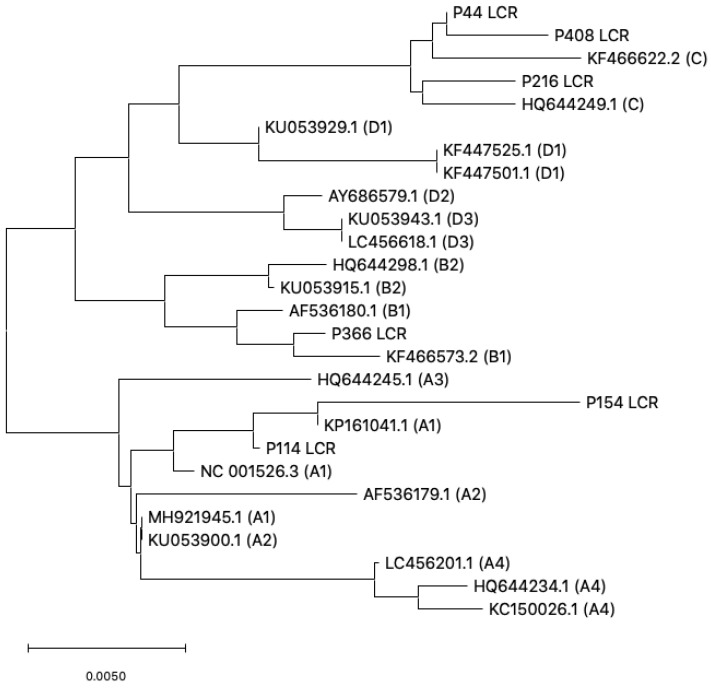
**Evolutionary relationships of HPV16 lineages/sublineages in Kinshasa based on LCR region.** The evolutionary history was inferred using the Neighbor-Joining method [37]. The tree is drawn to scale, with branch lengths in the same units as those of the evolutionary distances used to infer the phylogenetic tree. The evolutionary distances were computed using the Maximum Composite Likelihood method [38] and are in the units of the number of base substitutions per site. The analysis involved 27 nucleotide sequences (21 reference strains from GenBank and 6 new sequences from this study). Codon positions included were 1st + 2nd + 3rd + Noncoding. All positions containing gaps and missing data were eliminated. There was a total of 985 positions in the final dataset. Evolutionary analyses were conducted in MEGAX [35,36].

**Table 1 microorganisms-10-02492-t001:** Primer pairs for HPV16 L1 and LCR regions.

Target Region	Primer Name *	Sequence	Amplicon Size (bp)
L1	L1_16_1F	ATGTCTCTTTGGCTGCCTAG	911
L1	L1_16_1R	GCATCAGAGGTAACCATAGAAC	
L1	L1_16_2F	CTATGGACTTTACTACATTACAGGCTA	879
L1	L1_16_2R	GTTTAGCAGTTGTAGAGGTAGATGA	
LCR	LCR_1F	ACCCACCACCTCATCTACCTCTACAA	460
LCR	LCR_1R	ATTTGGCACGCATGGCAAGCAGGAA	
LCR	LCR_2F	CATGCTTTTTGGCACAAAATGTGTTTT	621
LCR	LCR_2R	ATATCATGTATAGTTGTTTGCAGCTCT	

* F: Forward; R: Reverse; Reference strain: NC_001526.

**Table 2 microorganisms-10-02492-t002:** L1 * nucleotide and amino acid substitutions by position and samples from Kinshasa (DRC).

Lineage	Sequence ID	Nucleotide Mutation Sites		Amino Acid Mutation Sites
A1	P114_L1	None		None
A1	P154_L1.2	A796G		T266A
C	P44_L1	G60A	C921T	H76Y
		C226T	A1057C	T176N
		T273C	G1083A	N181T
		C527A	C1332T	T266A
		A542C	C1216T	S282P
		T609C	C1227T	T353P
		A678G	G1356A	L474F
		A796G	G1422T	
		T844C		
C	P232_L1.2	A678G	G1083A	T266A
		A796G	C1216T	S282P
		T844C	C1227T	T353P
		C921T	C1332T	
		A1057C	G1356A	
C	P408_L1.2	A678G	C1216T	T266A
		A796G	C1227T	S282P
		T844C	C1332T	T353P
		C921T	G1356A	L474F
		A1057C	G1422T	
		G1083A		

* L1.1 covers positions 1 to 911 (aa 1–304). L1.2 covers positions 646 to 1518 (aa 207–499).

**Table 3 microorganisms-10-02492-t003:** LCR * nucleotide mutations by position and samples from Kinshasa (DRC).

Lineage	Sequence ID	Nucleotide Mutation Sites	
A1	P114_LCR	C7430T	
		C7876A	
A1	P154_LCR	G7193T	
		T7416A	
		G7521A	
		C7533G	
B1	P366_LCR	G7193T	C7764T
		A7232G	C7786T
		T7293G	A7797T
		G7489A	G7834T
		G7521A	C7876A
		C7689A	G28T
		T7714A	TT31C
C	P44_LCR	G7193T	C7764T
		A7233C	C7786T
		G7387C	A7826G
		G7435A	G7834T
		A7485C	A7837C
		G7489A	A7839G
		G7521A	T31C
		C7669T	T109C
		C7689A	
C	P216_LCR	G7435A	A7826G
		G7436C	G7834T
		A7485C	A7837C
		G7489A	A7839G
		G7521A	T31C
		C7669T	T109C
		C7689A	G132T
		T7700G	C143G
		C7764T	G145T
		C7786T	
C	P408_LCR	A7156C	C7786T
		A7159G	A7826G
		G7193T	G7834T
		A7233C	A7837C
		G7387C	A7839G
		G7435A	T31C
		A7485C	T109C
		G7489A	G122C
		G7521A	G124C
		C7669T	G132T
		C7689A	C143G
		C7764T	G145T

* LCR.1 covers positions 7101 to 7562. LCR.2 covers positions 7467 to 7906 and 1 to 180.

## Data Availability

The data presented in this study are openly available in Harvard Dataverse https://dataverse.harvard.edu/dataset.xhtml?persistentId=doi:10.7910/DVN/UWZWER (accessed on 23 November 2022).

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
