# Peer review of "Characterization of Human Papillomavirus 16 from Kinshasa (Democratic Republic of the Congo)—Implications for Pathogenicity and Vaccine Effectiveness"

_microorganisms, 2022, doi:10.3390/microorganisms10122492_

Round 1
Reviewer 1 Report
Paula Iglesias et al characterized the HPV types epidemiology in Kinshasa and prevalence of HPV 16 in LSIL and HSIL and the presence of genomic variants in L1 gene and LCR of samples positives for HPV 16 infection. They suggest that HPV 16 is the principal HPV associated with HSIL and that its genomic variation could contribute to explain the the vaccine-induced neutralization.
Some of the conclusions drawn are not supported by the data presented and I have some concerns regarding the results showed in this study.
Major Points:
1.- The authors the have some confusion regarding the structure of the HPV genome and the sequences that code for a functional product, such as viral proteins. In some sections they describe the LCR as if it were a gene, but it is a regulatory region and it is not coding. “Similarly, LCR was divided in two fragments of 460bp (LCR.1) and 621 bp (LCR.2) for a final complete LCR sequence of 985 bp (amino acids 2367-2365 and 1-60 in the reference sequence NC_001526.3).”.
This is also seen when they describe the genetic variants of the L1 and LCR regions, showing LCR mutations as mutations that cause amino acid changes (page 9/15) “Additionally, C7485A, C7669T, G7826A, A7837C and T109G were present in 3/6 samples, leading to amino acid substitutions K2495N, H2557Y, C2609Y, K2613Q andS37P, respectively.”.
2.- The authors show in figures 1, 2 and 3 the percentages of detection of the different HPV genotypes and the number of cases of infections by HPV 16, however, when adding the percentages that they present, in these three figures, they are above 100%. The authors must review their results again to make the pertinent corrections. For example in Figure 3, Prevalence of HPV among participants with high-grade squamous intraepithelial lesion (HSIL) in Kinshasa: 55.6+11.1+11.1+11.1+11.1+11.1 = 111.1%.
3.- The figures 4 and 5 phylogenetic tree of the HPV16 variants based on L1 and LCR sequences are confusing. I think they should describe how they were obtained. “Phylogenetic tree of the HPV16 variants based on L1 sequences of samples from Kinshasa (as per study ID) and samples sequences from GenBank database. There were a total of 1,518 positions in the final dataset of the tree”. Why and how the authors did this analysis?
4.- The authors mentioned that among HPV16 positive samples, L1 and LCR regions could be sequenced from 6 and 8 patients, respectively, a total of 14 samples. However, in section 3.1 the phylogenetic analysis allowed to obtain the lineage of 7 out of 11 HPV 16 detected among the screened woman. what happened to the other 3?
5.- In section 3.2 the authors say “Seventeen single nucleotide changes were identified among the five L1 sequences obtained.”, but they said before that L1 was sequencing in 6 patients. The number of samples do not match.
6. In section 3.3 it is the same situation. The authors analyzed the nucleotide changes in 6 LCR sequences, however, they said that they could sequence 8 patients.
7. The authors concluded that L1 mutations were frequent among HPV16 strains in Kinshasa and may affect the neutralization induced by vaccines, but the sample size is too small and they say that it is necessary a larger sample size, in this case, I considered that they could not conclude this effect over vaccination, moreover, they do not know the status of patient vaccination.
Minor points
I recommend to update some references such as reference 5, the epidemiology data is from 2010, the author could find some more recent data.
Author Response
Response to academic editor and reviewers
Ref. No.: Microorganisms-2036509
Title: Characterization of Human Papillomavirus 16 from Kinshasa (Democratic Republic of the Congo). Implications for pathogenicity and vaccine effectiveness.
I would like to thank the reviewers and the editor for the time and effort that they have put into evaluating the earlier version of the manuscript. Below you will find our point-by-point responses to the comments.
A new version of figure 1 is included in the manuscript without “track changes”.

Reviewer 2 Report
“Characterization of Human Papillomavirus 16 from Kinshasa (Democratic Republic of the Congo). Implications for pathogenicity and vaccine effectiveness”, the prevalence of HPV types and HPV16 intratype variants were analyzed in low- and high-grade squamous epithelial lesions of the cervix (LSIL/HSIL) in a population of Democratic Republic of the Congo (DRC).
In this study, the prevalence of HPV was determined in endocervical scrapes from 480 women from 4 outpatient centers of Kinshasa (DRC). Of women screened 18.8 % were positive for HPV, being 75% of high-risk types. The strains or intratype variants of HPV16, which was the most prevalent type (12.2%), were characterized, this analysis was performed in 11 cases that were positive for PH16
Phylogenetic trees of HPV16 were obtained from L1 and LCR sequences, where the different lineages were identified, prevailing the AFR2 C lineage. Overall, fifty-two single nucleotide changes were identified: seventeen in L1 (involving 10 aminoacid changes), and 35 in LCR (24 aminoacid changes). Those changes were fully characterized.
The authors are aware that a limitation of the study is the low number of samples (11) analyzed
The manuscript is in general well written and contains relevant information in the field. Since there is a lack of information on this regard in Africa, and particularly in CDR, this manuscript supports epidemiological studies on HPVs and their intratype variants which could eventually impact HPV vaccine design as well as local vaccination policies.
Comments:
Information on incidence and mortality of cervical cancer in DRC could be included in the introduction. As reported in Globocan 2020, IARC cervical cancer ranks in second place in incidence and mortality in DCR.
Author Response
Response to reviewer 2
Ref. No.: Microorganisms-2036509
Title: Characterization of Human Papillomavirus 16 from Kinshasa (Democratic Republic of the Congo). Implications for pathogenicity and vaccine effectiveness.
I would like to thank the reviewers and the editor for the time and effort that they have put into evaluating the earlier version of the manuscript. Below you will find our point-by-point responses to the comments.
A new version of figure 1 is included in the manuscript without “track changes”.

Reviewer 3 Report
Journal: Microorganisms (ISSN 2076-2607)
Title: Characterization of Human Papillomavirus 16 from Kinshasa (Democratic Republic of the Congo). Implications for pathogenicity and vaccine effectiveness.
Type: Research article
Manuscript ID: microorganisms-2036509
Authors: Paula Iglesias , Celine Tendobi , Silvia Carlos * , Maria D. Lozano , David Barquín , Luis Chiva , Gabriel Reina
Description: The present manuscript reports on an investigation of the identification of DNA sequences belonging to HPV in a group of endocervical cytology samples that were collected from 480 women attending a voluntary cervical cancer screening program at Monkole Hospital and four outpatient centers in Kinshasa (Democratic Republic of the Congo). While viral DNA has been identified, several HPV genotypes, including the oncogenic ones, mainly HPV16, have been molecularly characterized. Polymorphisms on the viral long control region have been identified, reported and discussed. Authors stated that the ms provides a helpful basis for understanding HPV16 variants circulating.
Revision: This is an interesting study evaluating for the first time the sequence variability across different HPV16 lineages is a specific African region. The ms is in general well written, while the experimental design is well performed. With some exceptions, figures are highly informative and clear
The authors may consider the following points for improving the manuscript.
1) Line 27 “(DRC)” should be “Democratic Republic of the Congo (DRC)” Please check the entire ms for additional inaccuracies regarding the acronyms. A similar observation can be made for LCR which should be reported as “Long control region (LCR)”.
2) Introduction, HPV16 is the main causative factor of cervical, vulvar, penile and anal cancers and of carcinomas of the upper respiratory tract. For Completeness of information, this notion should be included in.
3) Lines45-47 Besides oncogenic HPV activity, cervical cancer onset and development encompass a large variety of genetic and epigenetic changes. (DOI: 10.1002/jcp.24808 and DOI: 10.3389/fgene.2020.00347 ). This notion, alongside supporting references should be included.
4) Lines66-77 HPV long control region is under epigenetic regulation (doi: 10.3390/pathogens9060483)
5) Lines 78-82 Are these L1-based variants specific for HPV16?
6) In order to improve the readability of the work, I suggest removing the subhead titles in the discussion.
7) Please include references in the methods
8) DNA isolation procedures should be separated from PCR methods and moved above 2.4 section, while 2.4 section should be merged with the PCR methods.
9) The quality of fig 1 should be improved, as being difficult to read
10) Lines 112-113 the exact HPV genotypes detectable with the array should be included
11) Line 365 specific viral proteins regulated by LCR region should be mentioned
Author Response
Response to academic reviewer 3
Ref. No.: Microorganisms-2036509
Title: Characterization of Human Papillomavirus 16 from Kinshasa (Democratic Republic of the Congo). Implications for pathogenicity and vaccine effectiveness.
I would like to thank the reviewers and the editor for the time and effort that they have put into evaluating the earlier version of the manuscript. Below you will find our point-by-point responses to the comments.
A new version of figure 1 is included in the manuscript without “track changes”.

Round 2
Reviewer 1 Report
The authors present the study Characterization of Human Papillomavirus 16 from Kinshasa (Democratic Republic of the Congo). Implications for pathogenicity and vaccine effectiveness.
In the first review I made some observations that were attended and corrected.
I have no more comments for the authors.